biological applications, ecology, physiology

ecological light pollution, artificial lightscapes, anthropogenic stressor, mortality, pollutant

**Author for correspondence:**
Suzanne C. Mills
e-mail: suzanne.c.mills@gmail.com

†Co-first author.
‡Present address: Institute of Biodiversity, Animal Health and Comparative Medicine, University of Glasgow, Glasgow, UK.

# Long-term exposure to artificial light at night in the wild decreases survival and growth of a coral reef fish

Jules Schligler[1,†], Daphne Cortese[1,†,‡], Ricardo Beldade[1,2], Stephen E. Swearer[3] and Suzanne C. Mills[1,4]

[1]USR 3278 CRIOBE, BP 1013, PSL Université Paris: EPHE-UPVD-CNRS, 98729 Papetoai, Moorea, French Polynesia
[2]Las Cruces, Pontificia Universidad Católica de Chile, Estación Costera de Investigaciones Marinas and Center for Advanced Studies in Ecology and Biodiversity, Santiago de Chile, Chile
[3]National Centre for Coasts and Climate and School of BioSciences, University of Melbourne, Parkville, Victoria, 3010, Australia
[4]Laboratoire d'Excellence 'CORAIL', France

JS, 0000-0001-7695-8790; DC, 0000-0002-5746-3378; RB, 0000-0003-1911-0122; SES, 0000-0001-6381-9943; SCM, 0000-0001-8948-3384

Artificial light at night (ALAN) is an increasing anthropogenic pollutant, closely associated with human population density, and now well recognized in both terrestrial and aquatic environments. However, we have a relatively poor understanding of the effects of ALAN in the marine realm. Here, we carried out a field experiment in the coral reef lagoon of Moorea, French Polynesia, to investigate the effects of long-term exposure (18–23 months) to chronic light pollution at night on the survival and growth of wild juvenile orange-fin anemonefish, *Amphiprion chrysopterus*. Long-term exposure to environmentally relevant underwater illuminance (mean: 4.3 lux), reduced survival (mean: 36%) and growth (mean: 44%) of juvenile anemonefish compared to that of juveniles exposed to natural moonlight underwater (mean: 0.03 lux). Our study carried out in an ecologically realistic situation in which the direct effects of artificial lighting on juvenile anemonefish are combined with the indirect consequences of artificial lighting on other species, such as their competitors, predators, and prey, revealed the negative impacts of ALAN on life-history traits. Not only are there immediate impacts of ALAN on mortality, but the decreased growth of surviving individuals may also have considerable fitness consequences later in life. Future studies examining the mechanisms behind these findings are vital to understand how organisms can cope and survive in nature under this globally increasing pollutant.

## 1. Introduction

Artificial light at night (ALAN) is a globally widespread environmental pollutant with direct ecological impacts on multiple terrestrial and aquatic ecosystems [1–3] including ecosystem functioning [4]. ALAN has been identified as a new major pollutant in the context of global environmental change [1,5]. Levels of light pollution are closely associated with human population density and economic activity [6]. Approximately one-tenth of the world's population (600 million people) live in coastal areas that are less than 10 m above sea level, resulting in considerable anthropogenic light pollution [7], which is expected to increase in parallel with global human population increases along the world's coastline [8]. Light pollution is a recognized threat for wildlife and biodiversity worldwide [9,10], directly affecting biological and ecological processes across taxa, including changes in key life-history traits, such as immune function [11,12], survival [13], ageing [14], and fecundity [15], however, the impacts of ALAN have rarely been assessed for marine species in the wild.

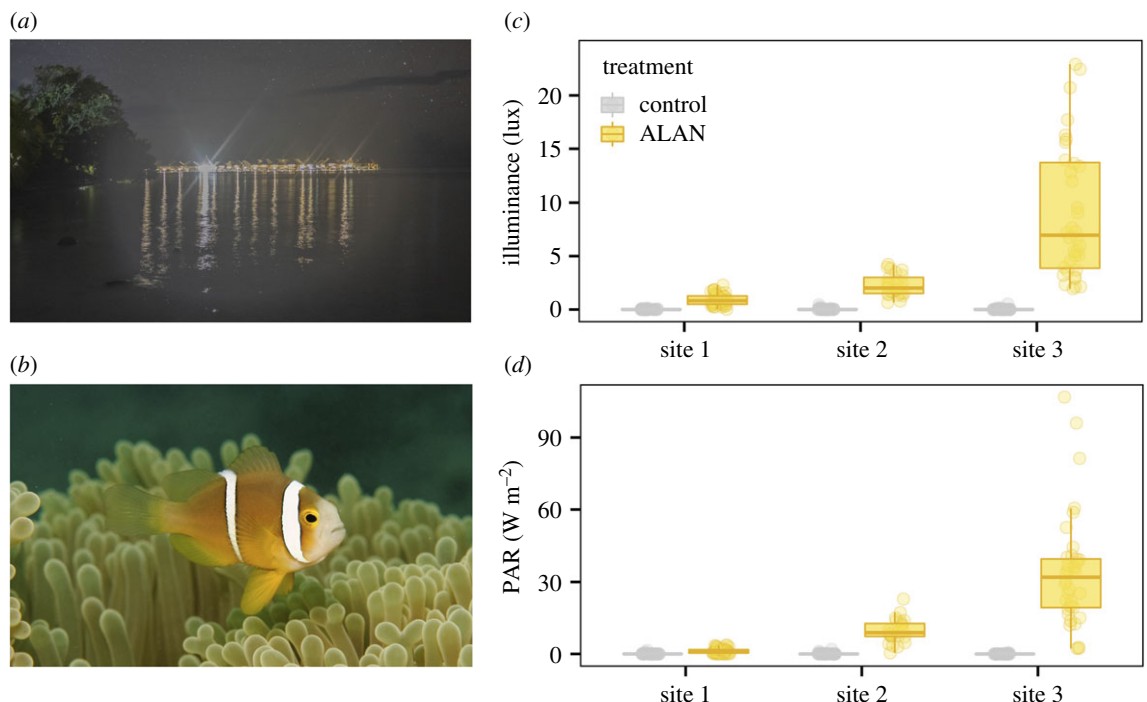

**Figure 1.** Photographs of (*a*) artificial light at night (ALAN) in Moorea, French Polynesia (photo credit: Jules Schligler) and (*b*) an orange-fin anemonefish, *Amphiprion chrysopterus*, in its host anemone *Heteractis magnifica* (photo credit: Fred Zuberer), and underwater light intensity measurements in (*c*) illuminance (lux) and (*d*) photosynthetic active radiation (PAR) in both treatments at each of the three sites. (Online version in colour.)

Organisms have evolved biological rhythms, and light cycles are the strongest and most predictable of environmental cues, such that cues from circadian, circalunar, seasonal, and annual rhythms are vital for biological processes such as reproduction, migration, and dispersal [9,16,17]. Natural light regimes may be compromised by anthropogenic night-time light pollution, directly disrupting physiology, behaviour, and reproduction in many organisms [18]. ALAN may also alter interactions with other species, on the one hand by facilitating foraging and energy acquisition [19,20], but also by increasing vulnerability to predators [21]. It is therefore unsurprising that there is little consensus as to ALAN's effects on an organism's survival and growth. The studies comparing continuous exposure to light have produced conflicting results, ranging from increased growth [22,23], no difference in growth [24], variation in growth responses across species [25], and decreased growth [26,27]. Furthermore, exposure to ALAN has been shown to increase animal mortality in some species [13,28], but not in others [15]. The majority of these studies were conducted under laboratory conditions, which may not adequately portray the effects of ALAN on survival and growth in natural populations [5] where animals interact with predators and prey, and where habitat complexity and coping behaviours may hamper detrimental effects of ALAN.

We investigated the impact of long-term exposure of ALAN on the survival and growth of resident juvenile orange-fin anemonefish, *Amphiprion chrysopterus* (Cuvier, 1830) in the lagoon of Moorea, French Polynesia. Anemonefish live in an obligate mutualism with sea anemones, mainly with the magnificent sea anemone, *Heteractis magnifica*, making them an ideal model species for long-term *in situ* studies [29–31]. Anemonefish actively feed on plankton in the water column [32] and on anemone waste products [33]. Sea anemones also provide anemonefish and their eggs shelter from predators [32]. Anemonefish are restricted to living near their host anemone and thus are unable to move away from the direct effects of a potential stressor such as light or sound pollution [29,30]. Light pollution may also indirectly impact anemonefish via its direct impacts on their planktonic invertebrate prey [34], anemones [35], or predators [36].

In a field-based experiment, we aimed to determine the impacts of ALAN on the survival and growth of juvenile orange-fin anemonefish, a species with an average lifespan of approximately 14 years (R Beldade 2021, personal communication), over, on average, 20.5 months. We predicted that the long-term exposure to light pollution would (i) lower survival due to the potential of light to attract natural predators [36] and (ii) increase anemonefish growth due to the increased availability of planktonic prey that are attracted to light [34,37], coupled with the potential for extended daily foraging [38] and increasing foraging rates [39] by these diurnal zooplanktivores [32]. However, if such advantages trade-off with any direct physiological costs of ALAN and sleep deprivation [40], (iii) light pollution would have no impact on growth. We specifically tested our predictions on juvenile anemonefish given the high mortality [41] and greater potential for developmental acclimation e.g. [42], during this life stage.

## 2. Methods

### (a) Experimental sites

We surveyed three subtidal fringing reef sites exposed to long-term terrestrial lighting (ALAN) along the shores of Moorea, French Polynesia (17°32′19.8″ S, 149°49′46.3″ W; figure 1*a*; electronic supplementary material, figure S1). We paired each of the three ALAN sites with a nearby control site of comparable depth and reef structure, but with no artificial illumination and only moonlight (and potential skyglow) (distance between treatments—control and ALAN territories: site 1 = 8.6 ± 1 m; site 2 = 35 ± 1 m; site 3 = 403 ± 1.5 m; electronic supplementary material, figure S1). We paired the ALAN site at site 2 with a second control site with a more similar reef structure (distance between treatments =

610 ± 2.9 m; electronic supplementary material, figure S1). We created six anemonefish territories at each control and ALAN site (42 territories in total). Each territory consisted of one healthy magnificent sea anemone, *Heteractis magnifica*, which had been collected in the northern lagoon of Moorea and randomly distributed across the 42 territories. We measured light intensities at the control and ALAN sites underwater as illuminance (in lux; lumen per square metre, a measure of the intensity of light as perceived by the human eye) and photosynthetically active radiation (PAR in W m$^{-2}$; the amount of light available for photosynthesis) at the same depth as anemones (60 cm) using a SpectroSens2+ (Skye Instruments Ltd). We took measurements in triplicate at one location in each of the six sites at night after moonrise, weekly (corresponding to the four lunar phases) on 12 different nights over four months (216 measurements in total, 36 per site). Distances of anemone territories from the coast varied between sites and treatments (site 1: control = 127 ± 23 m, ALAN = 29 ± 3 m; site 2: controlA = 46 ± 3 m, controlB = 322 ± 3 m, ALAN = 28 ± 3 m; site 3: control = 89 ± 3 m, ALAN = 17 ± 1 m; electronic supplementary material, figure S1).

## (b) Anemonefish monitoring

We used wild-spawned, laboratory-reared, juvenile orange-fin anemonefish, *Amphiprion chrysopterus*, of one to three months of age from approximately 50 wild breeding pairs in the experiment. We collected eggs from natural nests in Moorea lagoon over three months which were hatched and reared in aquaria at CRIOBE (electronic supplementary material S1). After approximately 1–3 months, we photographed and weighed (±0.01 g; Ohaus Scout Pro Portable Electronic Balance) 42 juveniles. We measured total length (TL: distance from tip of lower jaw to end of tail) and body height (distance from the pelvic fin and the start of the dorsal fin) (±0.001 cm) from photographs using ImageJ software. We randomly placed individual juvenile anemonefish into each experimental anemone, either control ($n = 24$) or ALAN ($n = 18$) with no difference in length, height, or weight between treatments (electronic supplementary material S2). We released juveniles within approximately 10 cm of the anemone and they always swam straight into the anemone (figure 1b).

Juvenile anemonefish were exposed to their respective treatments for a mean duration of 20.5 months (mean exposure (±s.e.): control = 20 months and eight days (±13 days); ALAN = 20 months and 20 days (±13 days)). The developmental life stages in anemonefish are determined by gonadal development and encompass juveniles, immature sub-adults, and functional males and females. The timing of developmental stages is not known for *A. chrysopterus*, but juvenile saddleback anemonefish, *A. polymnus*, become immature sub-adults after 2–4 months and sex inversion into a functional female occurs between 12–14 months [43]. To determine whether exposure to ALAN was more sensitive across each transition period, we aimed to monitor survival prior to and after these developmental stages. We therefore monitored the survival of juvenile *A. chrysopterus* on four occasions: after 35 days (approx. one month as juveniles), 70 days (approx. two months as immature sub-adults), 12–17 months (mean: 15 months prior to or during sex inversion), and 18–23 months (mean: 20.5 months as functional adults). At each monitoring period, we visited each of the 42 anemone territories from shore (all three sites), except the second set of control territories at site 2 which we visited by boat. We monitored the presence of an anemone and an anemonefish. We then captured the fish using hand nets and photographed them to confirm individual identification.

## (c) Survival

We determined anemonefish survival (absence/presence), as well as their host anemone survival (absence/presence), during the four monitoring periods. Even though the anemonefish were able to move, the distance between anemones within treatment (mean ± s.e.; control = 5.3 ± 0.1 m, ALAN = 4.1 ± 0.2 m) and especially between treatments (overall mean = 264 ± 53 m; see the section on experimental sites for more details), and the high predation risk outside the anemone tentacles [32,44], especially for juveniles [45], reduced the chance of fish movement. Nevertheless, a previous study testing movement behaviour in six anemonefish species found that adult and juvenile *A. chrysopterus* were the most likely to move (13% changed host anemone within four months compared to 6% in *A. clarkii* and 0% in four other *Amphiprion* species)[46]. Therefore, to confirm individual identification, we photographed focal fish at each monitoring period and identified individuals by colour patterns based on the shape and length of the second and third vertical white stripes (electronic supplementary material, figure S2). The absence of a fish from its anemone was equated to mortality, which is particularly high upon settlement [44,46]. The exact date of death (absence) is unknown; therefore, we determined individual survival times as the average survival (in months) between the maximum survival time (i.e. date of most recent monitoring) and the minimum survival (i.e. date of the preceding monitoring). If both the anemonefish and anemone were absent, individual anemonefish were removed from the survival analysis (three control and four ALAN) since the cause of fish absence is not known (either a direct treatment effect or indirectly from the disappearance of the host anemone [47]). An equal number of anemones from both treatments were absent at the end of the experiment (seven control = 29.1%, seven ALAN = 38.8%), i.e. four control and three ALAN anemonefish changed host, and were included in the survival analysis, but we have no information whether they moved before or after the anemone disappeared.

## (d) Specific growth rate

At the end of the last two exposure periods (15 and 20.5 months), we re-captured, re-weighed, and re-photographed all surviving individuals (electronic supplementary material, figure S2). We determined specific growth rate (SGR) as the percentage increase in individual size (body mass, height, or total length) per day as $SGR = \ln(S_{t_2}) - \ln(S_{t_1}) \times t^{-1} \times 100$, where $S$ is the fish size at $t_2$ (final time) and $t_1$ (initial time) and $t$ is the time (days) between the two consecutive measures [48].

## (e) Statistical analyses

We performed statistical analyses in R v. 4.0.4 [49]. We determined differences in illuminance (lux) and PAR ($\log 10(x + 1)$-transformed) among sites and treatments (categorical variables) using linear mixed-effect models (LMERs), and replicate was added as a random effect with lunar phase and month as covariates. We fitted LMERs using the *lmerTest* package [50], while marginal (m$R^2$) and conditional (c$R^2$) $R^2$ were obtained with the package *piecewiseSEM* [51]. In addition, we determined differences in maximum illuminance (lux) and PAR among sites and treatments using linear models (LM).

We determined the effect of treatment and site on the probability of fish survival over time using a general linear model (GLM) with a binomial distribution. We fitted survival curves using the *survminer* package [52].

We fitted LMERs to analyse the effect of treatment, site, period of treatment exposure (0–15 versus 15–20.5 months), and initial fish height, length, and weight (log-transformed) on fish SGR (height, length, and log-transformed weight).

We determined best-fit models using likelihood ratio tests, starting from the most complex model and subsequently removing non-significant interactions and explanatory variables via the *lmtest* package [53], and we used Tukey *post-hoc* tests to determine differences among groups via the 'emmeans' function in the *emmeans* package [54].

# 3. Results

## (a) Light intensities

Mean light intensities (in lux) at ALAN territories were, on average, 143 times higher than intensities at control territories at all sites (electronic supplementary material, table S1A–C; *post-hoc* test on LMER – site 1 (control versus ALAN): $t = -4.865$, $p < 0.001$; site 2: $t = -6.140$, $p < 0.001$; site 3: $t = -10.164$; $p < 0.001$; figure 1*c*) and similarly for PAR (electronic supplementary material, table S2A–C; site 1: $t = -3.690$, $p = 0.001$; site 2: $t = -6.128$, $p < 0.001$; site 3: $t = -8.218$; $p < 0.001$; figure 1*d*). The light intensities at each of the three ALAN sites (in lux and PAR) were significantly different from each other, with site 3 > site 2 > site 1 (all $p < 0.01$; electronic supplementary material, tables S1D and S2D; figure 1*c,d*). However, there were no differences in light intensities among control sites (all $p > 0.92$; electronic supplementary material, tables S1D and S2D; figure 1*c,d*). The variability observed within each site was due to differences among triplicate measures (electronic supplementary material, tables S2A and S2B), rather than lunar phase or month (electronic supplementary material, tables S1A and S2A). The same results were found if we only used the maximum of the triplicate light intensity measures (in lux and in PAR) (electronic supplementary material, tables S3A–D and S4A–D).

## (b) Survival

Only approximately half of the anemonefish were present in their host anemones 12–17 months later (15 controls, six ALAN, and one new recruit on an ALAN territory) and only one-third 18–23 months later (12 controls and three ALAN). The survival probability of fish exposed to ALAN tends to be lower (36%) than that of control individuals exposed to natural light at night (GLM: treatment (ALAN); $n = 35$; $z$-value = 1.921; $p = 0.055$; electronic supplementary material, table S5B, figure 2). The survival probability of anemonefish decreased over time in a similar manner between the treatments except during two periods where anemonefish exposed to ALAN had lower survival: prior to the first monitoring period (juvenile stage) and after approx. 20 months of exposure (as functional adults) (figure 2). In addition, differences were observed among sites, with fish from site 2 having 51% and 55% higher survival respectively, compared to fish from site 1 ($z$-value = 0.416; $p = 0.047$; electronic supplementary material, table S5C) and site 3 ($z$-value = 0.303; $p = 0.045$; electronic supplementary material, table S5C).

## (c) Specific growth rate

Growth was measured for 21 individuals (15 controls and six ALAN) after 12–17 months, and 15 individuals (12 controls and three ALAN) after 18–23 months. The SGR of juvenile anemonefish height was 15% lower (adjusted to initial size) in the ALAN treatment compared to the control (LMER: treatment (ALAN); $t = -2.260$, $p = 0.036$, $mR^2 = 0.906$, $cR^2 = 0.963$, table 1*a*; electronic supplementary material, table S6B; figure 3*a*), and they tended to have a 51% lower SGR in terms of weight (LMER: $t = -1.850$, $p = 0.074$, $mR^2 = 0.397$, $cR^2 = 0.397$, table 1*c*; electronic supplementary material, table S8B, figure 3*c*). Despite a 21% lower SGR in terms of length, we found no statistical difference (LMER: $t = -1.184$, $p = 0.251$, $mR^2 = 0.846$, $cR^2 = 0.888$, table 1*b*; electronic

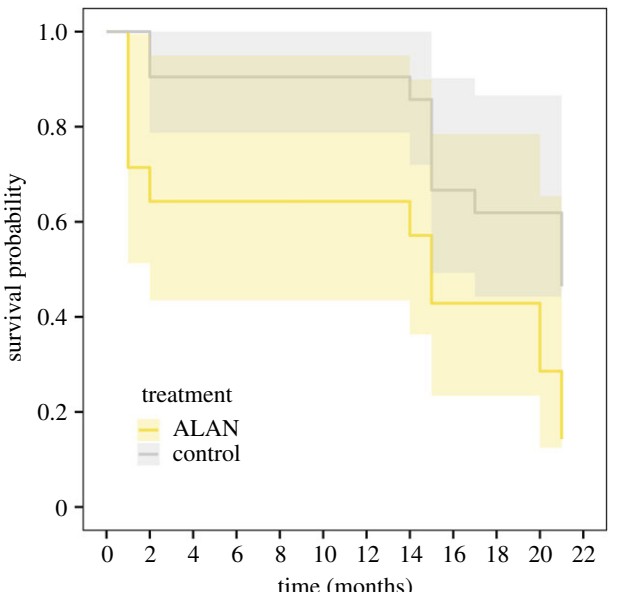

**Figure 2.** Survival rate with 95% confidence bands of juvenile orange-fin anemonefish, *Amphiprion chrysopterus*, exposed to either control (natural light cycle) or ALAN over a maximum of 23 months. Since the exact date of death (absence) is unknown, the graph is based on the average of individual survival time calculated as the mean between the maximum survival time (i.e. date of most recent monitoring) and the minimum survival (i.e. date of the preceding monitoring). (Online version in colour.)

supplementary material, table S7B; figure 3*b*). We did not observe any interaction among treatment and time of exposure (electronic supplementary material, tables S6A, S7A, and S8A). Overall, daily growth rate was higher during the second period of exposure than in the first (LMER: height (2nd exposure period); $t = 3.752$, $p < 0.001$, length (2nd exposure period); $t = 3.796$, $p < 0.001$, weight (2nd exposure period); $t = 1.914$, $p = 0.065$).

We did not observe a difference in juvenile growth among sites (all $p > 0.05$: electronic supplementary material, tables S6B, S7B, S8B); however, we did observe an interaction between exposure duration and site on growth in height with fish from site 2 growing more than those from site 3, but only in the first exposure period (electronic supplementary material, table S6C; *post-hoc* test on LMER—1st exposure period (site 2 versus site 3); $t = 3.241$, $p = 0.009$). As expected, given that daily growth declines with size [55], initial fish height, length, and body mass had a negative effect on growth rate (LMER: initial height; $t = -16.490$, $p < 0.001$, electronic supplementary material, table S6B; LMER: initial length; $t = -11.728$, $p < 0.001$, electronic supplementary material, table S7B; LMER: initial weight; $t = -3.723$, $p < 0.001$; electronic supplementary material, table S7B).

# 4. Discussion

Our study provides the first evidence that long-term exposure to ALAN pollution, over 18–23 months, negatively impacts the survival and growth of a wild coral reef fish. Our results present a comprehensive understanding of the impact of ALAN on survival and growth in the natural environment simultaneously considering the focal fish, their predators and prey. The levels of illuminance measured underwater at our three ALAN sites, 0.9, 2.3, and 8.9 lux, are representative

**Table 1.** Results of the best-fit linear mixed-effect model performed on (a) specific growth rate (SGR) in height, (b) SGR in length, and (c) SGR in weight as response variables. Model selection is reported in electronic supplementary material, tables S5A, S6A, and S7A. $H_i$, $L_i$, and $W_i$ are the initial fish measurements at the start of the experiment, treatment is the light treatment (control versus ALAN), and site is field site. Exposure period refers to the two periods of treatment exposure (i.e. 0–15 months and 15–20.5 months). $mR^2$ is the marginal $R^2$, which describes the proportion of variance explained by the fixed factors alone. $cR^2$ is the conditional $R^2$, which describes the proportion of variance explained by both fixed and random factors. Significant data are shown in italics.

| fixed effects | estimate | s.e. | d.f. | t-value | p-value | $mR^2$ | $cR^2$ |
|---|---|---|---|---|---|---|---|
| (a) SGR in height | | | | | | | |
| *intercept* | *0.2641* | *0.019* | *24.52* | *13.472* | *<0.001* | 0.906 | 0.963 |
| *log($H_i$)* | *−0.2172* | *0.013* | *25.22* | *−16.490* | *<0.001* | | |
| *treatment (ALAN)* | *−0.0309* | *0.014* | *18.21* | *−2.260* | *0.036* | | |
| *2nd exposure period* | *0.0758* | *0.020* | *17.09* | *3.752* | *<0.001* | | |
| site 2 | 0.0325 | 0.019 | 22.52 | 1.691 | 0.105 | | |
| site 3 | −0.0263 | 0.023 | 23.31 | −1.162 | 0.257 | | |
| *2nd exposure period * site 2* | *−0.0433* | *0.019* | *13.92* | *−2.169* | *0.048* | | |
| 2nd exposure period * site 3 | −0.0168 | 0.025 | 14.03 | −0.661 | 0.519 | | |
| (b) SGR in length | | | | | | | |
| *intercept* | *0.4671* | *0.033* | *25.90* | *14.093* | *<0.001* | 0.846 | 0.888 |
| *log($L_i$)* | *−0.2050* | *0.017* | *29.08* | *−11.728* | *<0.001* | | |
| treatment (ALAN) | −0.0161 | 0.014 | 19.23 | −1.184 | 0.251 | | |
| *exposure period* | *0.0515* | *0.014* | *26.66* | *3.796* | *<0.001* | | |
| site 2 | 0.0079 | 0.017 | 18.20 | 0.455 | 0.655 | | |
| site 3 | −0.0333 | 0.021 | 18.78 | −1.569 | 0.133 | | |
| (c) SGR in weight | | | | | | | |
| intercept | −0.3961 | 0.597 | 30.0 | −0.664 | 0.512 | 0.397 | 0.397 |
| *log($W_i$)* | *−0.7070* | *0.189* | *30.0* | *−3.723* | *<0.001* | | |
| treatment (ALAN) | −0.7874 | 0.426 | 30.0 | −1.850 | 0.074 | | |
| exposure period | 0.8427 | 0.440 | 30.0 | 1.914 | 0.065 | | |
| site 2 | −0.3848 | 0.541 | 30.0 | −0.722 | 0.482 | | |
| site 3 | −0.2458 | 0.668 | 30.0 | −0.368 | 0.716 | | |

of light pollution that shallow fringing reefs are exposed to under street lighting and hotel lights. As these levels are considerably lower than those produced by LED lights used at ports [56], the impacts of ALAN in coastal marine ecosystems could be greater than observed in the present study.

Our results agree with our prediction of lower survival in the ALAN treatment due to the potential for light to attract natural predators and/or its likely negative effects on physiology. Over the total monitoring period of 23 months, the survival probability of fish exposed to ALAN decreased by 36% compared to the control group (figure 2). The juvenile life stage of anemonefish was most affected by ALAN with high mortality during the first month of exposure. Juvenile survival in the wild was low, but this is comparable to levels previously observed for wild juvenile anemonefish [47]. Our finding of higher mortality in the wild may have been due to increased predation under ALAN as found for larvae of the coral reef fish, the convict tang, *Acanthurus triostegus*, in a laboratory study [57], and previous studies in the wild have demonstrated that the abundance of large predatory fish increases when exposed to artificial light pollution, which remain close to light [36]. Visual piscivorous predators may increase their activity under ALAN and enhance their foraging ability [36]. Sea anemone hosts provide a natural refuge from predation [32], and they may have partially mitigated the impact of increased predation and hence mortality. As such ALAN may have greater impacts on fish species, other than anemonefish, that are not associated with refuges from predation. Furthermore, hiding in anemones for long periods to avoid the increased presence of predators near lit sites may have hindered anemonefish feeding rates. If feeding was decreased under continuous light exposure, then anemonefish growth may have traded-off against survival and may explain the observed decrease in growth under ALAN. Lower survival may also be due to physiological stress as laboratory studies showed that ALAN decreased the survival of *A. triostegus* larvae in the absence of predators [57], caused a 12% reduction in the lifespan of the fruit fly *Drosophila melanogaster* [13,28], and ALAN decreased the condition of wild bird nestlings [58]. However, our results do not allow us to differentiate between physiological stress and predation to determine the cause of higher mortality in the wild under ALAN. Future studies should measure physiological traits and monitor the nocturnal behaviour of anemonefish, and other less site-attached species, as well as their predators, under ALAN to determine the cause of mortality.

Contrary to our predictions, as well as the findings from a laboratory study on coral reef fish larvae, *A. triostegus* [57], we

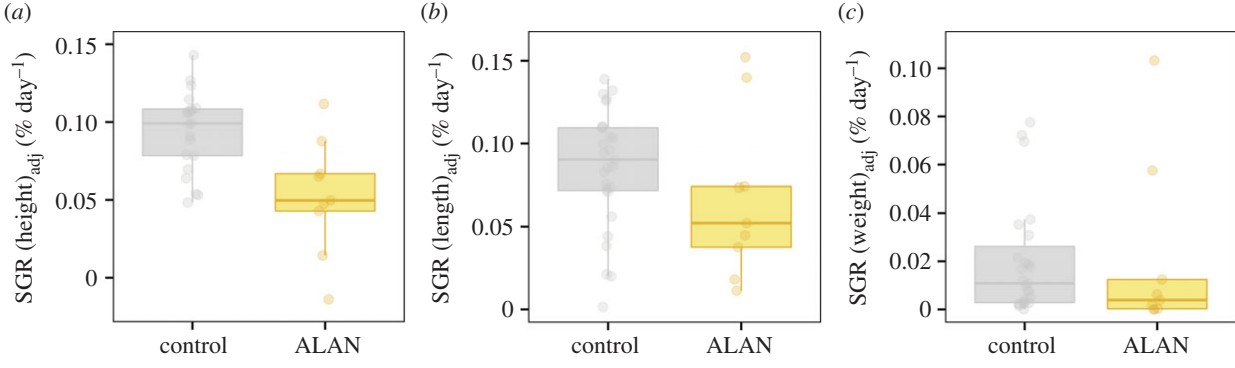

**Figure 3.** SGR of individual orange-fin anemonefish, *Amphiprion chrysopterus* (each data point), over both monitoring periods if alive (but repeated measures were accounted for in the model) for (*a*) height, (*b*) total length, and (*c*) weight. Individual data points were adjusted (using model residuals) to the overall mean measure of the fish size used in the growth analyses: (*a*) height (2.7 cm), (*b*) length (7.1 cm), and (*c*) body mass (11.0 g). (Online version in colour.)

observed a negative, rather than a positive, impact of ALAN on growth which was 15%, 21%, and 51% lower in terms of height, length, and weight, respectively. Growth is a complex process influenced by many interacting physiological and ecological factors and their trade-offs [55,59] and the observed lower growth may be due to negative effects on growth-related physiology. One such effect may be the requirement of a period of inactivity for optimal growth. The growth of the cinnamon anemonefish, *Amphiprion melanopus*, slowed when exposed to continuous lighting in aquaria [60] and the lack of darkness in our study may also have compromised growth in a similar manner. The lack of sleep may also result in increased metabolism, with a subsequent higher energetic demand [61], which despite the higher availability of planktonic prey attracted to light [34,37] coupled with the potential for extended daily foraging [38], might not be fully realized, especially if anemonefish hid in their anemones and actually fed less, resulting in less energy available for growth. The currently unknown impacts of ALAN on nocturnal fish behaviour in the wild, particularly foraging, may explain our contrasting results with those from a laboratory study where food was provided *ad libitum* to *A. triostegus* and larval growth was higher under ALAN [57]. Such contrasting results highlight the need for experimental manipulations of anthropogenic stressors to be carried out in the natural environment. The initial size of individual fish impacted growth, this is likely due to the fact that fish growth is nonlinear and asymptotic, so larger fish will grow less than smaller fish over the same period of time [55].

The physiological responses of juveniles could have carry-over effects on fitness-related traits later in life. Exposure to ALAN during early life stages that lower growth could result in a smaller size at maturity rendering these small fish more likely to lose in the competition for space and be more vulnerable to predation [62,63]. In addition, a poor start in life might further compromise adult reproduction that may already be negatively impacted by ALAN, as demonstrated by reduced egg hatching in a short-term laboratory study [28] with potential repercussions for population dynamics [64–66]. If the decrease in growth is sustained throughout life, it is likely that fish will be smaller at breeding, which, given the known positive size/fecundity relationship [67], and the importance of anemonefish size and rank for reproductive success [68] means that fish impacted by light will be less fecund and have lower reproductive fitness in adulthood [65].

Alternatively, the differences in growth may be due to phenotypic plasticity in morphology in response to ALAN and may be adaptive. ALAN may have induced a phenotypic change in resource allocation, giving priority to growth in length (and weight) over body height, representing a different growth strategy. For example, the morphology of crucian carp, *Carassius carassius*, has evolved differently in the presence/absence of predators to adaptively avoid predation [69]. *Amphiprion percula* show phenotypic plasticity in growth as juveniles [70] and indeed throughout life [71], and in this study, a lower growth rate, resulting in smaller height, may therefore be the optimal strategy for anemonefish exposed to ALAN and may facilitate escaping predation by either decreasing drag or hiding inside anemones. However, whether such plasticity is adaptive in the long-term remains unknown.

The growth of juvenile *A. percula* in the laboratory is under social control, whereby paired recruits grow more than solitary recruits [70] and in the wild *A. percula* settling into saturated anemones (large fish group size in relation to anemone size) suffer mortality after eviction by larger residents [44]. Our study may therefore be underestimating juvenile growth and mortality as a single juvenile anemonefish was placed in an empty anemone. However, our experimental design is representative of natural *Amphiprion* sp. recruitment which is higher into anemones with a low degree of saturation [44,46] and *A. chrysopterus* in French Polynesia live in small social groups, at small population densities and the population at the study site is recruitment-limited with many empty anemones [65,72,73]. It is not known to what extent growth and mortality is socially controlled in *A. chrysopterus* juveniles, but as our study design was the same for both treatments, any impact of the lack of conspecifics on growth and mortality would be similar for both treatments.

The impacts of other anthropogenic stressors, in particular those associated with near shore environments, such as runoff, especially nitrogen pollution [74], human disturbance from tourism [75], and land-based sound pollution [76], in addition to other marine stressors such as motorboat noise [30,77,78], may also have impacted the survival and growth of both anemonefish and their anemone hosts. However, it is unlikely that our results are confounded by anthropogenic stressors other than ALAN, as our experimental design specifically incorporated both treatments across three sites around Moorea differing in their exposure to each of the above anthropogenic stressors, except ALAN. Future studies

should quantify other anthropogenic stressors, in particular nitrogen pollution [74], in combination with ALAN to determine the combined interaction of these stressors.

In conclusion, combining the growing number of studies showing the negative impacts of ALAN in marine ecosystems with the projections of global population increases, especially along coastlines and the close association with levels of light pollution and population density, ALAN is already a risk to our marine ecosystems and will only exacerbate in the future. Marine-protected areas (MPAs) are not excluded from ALAN and due to the current lack of legislation, 20% of MPAs are already exposed to ALAN and 14.7% are exposed to increasing levels of light pollution [79], therefore mitigation measures should be of paramount importance. Mitigation measures and policy changes are urgently needed including maintaining and creating dark areas, only lighting part of the night and improving lighting technology in terms of directing light where it is needed, reducing light intensities, and changing spectra [80]. There is also growing concern regarding the combined interactions of multiple anthropogenic stressors, such as light and sound pollution [29,30] and the worldwide impact of these cumulative stressors needs to be better understood to help future management strategies [81].

Ethics. Ethical approval for the study was granted from The Animal Ethics Committee, Centre National de la Recherche Scientifique (permit no. 006725).

Data accessibility. Data available from the Dryad Digital Repository: https://doi.org/10.5061/dryad.37pvmcvjv [82].

Authors' contributions. J.S.: data curation, formal analysis, investigation, methodology, writing-original draft, writing-review, and editing; D.C.: data curation, formal analysis, investigation, methodology, writing-original draft, writing-review, and editing; R.B.: conceptualization, investigation, supervision, writing-original draft, writing-review, and editing; SE.S.: methodology, writing-review, and editing; S.C.M.: conceptualization, funding acquisition, investigation, methodology, project administration, resources, supervision, visualization, writing-original draft, writing-review, and editing

All authors gave final approval for publication and agreed to be held accountable for the work performed therein.

Competing interests. We declare we have no competing interests.

Funding. Financial support was provided by the Agence National de la Recherche (ANR-14-CE02-0005-01/Stay or Go) to G.A., S.C.M., and R.B., by the Haut-Commissariat de la République en Polynésie française (HC/3041/DIE/BPT/) to S.C.M. and Pacific Funds (BLEACH & ALAN) to S.C.M.

Acknowledgements. We are enormously grateful to Camille Vizon, Marie-Louise Elian, and Ignacio Pita Vaca for light intensity measures at night and for support in the lab and in the field during data collection; Anne Haguenauer, Frederic Zuberer, Guillaume Iwankow, Frank Lerouvreur, and Mathieu Kerneur for help in the field; Till Deuss and Robin Mannion for anemonefish larval rearing. The Underwater LUX Sensor and SpectroSens2+ handheld logging metre was kindly donated to S.C.M. by Skye Instruments Ltd (http://www.skyeinstruments.com/coral-research-and-skye-instruments/). We wish to thank Sofitel Moorea Ia Ora Beach Resort, Manava Beach Resort & Spa Moorea, and Intercontinental Moorea Resort & Spa for their help and support in the field.

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
