## [Peer Review File · Proceedings of the Royal Society B: Biological Sciences]

Review History

RSPB-2021-0454.R0 (Original submission)

Review form: Reviewer 1

Recommendation

Accept with minor revision (please list in comments)

Scientific importance: Is the manuscript an original and important contribution to its field?

Excellent

General interest: Is the paper of sufficient general interest?

Excellent

Quality of the paper: Is the overall quality of the paper suitable?

Excellent

Is the length of the paper justified?

Yes

Should the paper be seen by a specialist statistical reviewer?

No

Do you have any concerns about statistical analyses in this paper? If so, please specify them explicitly in your report.

No

It is a condition of publication that authors make their supporting data, code and materials available - either as supplementary material or hosted in an external repository. Please rate, if applicable, the supporting data on the following criteria.

Is it accessible?

N/A

Is it clear?

N/A

Is it adequate?

N/A

Do you have any ethical concerns with this paper?

No

Comments to the Author

This novel study uses a field-based experiment to test the effects of artificial light at night (ALAN) on the growth and survival of juvenile anemonefishes over a ~20 month period. Juvenile anemonefish from wild-spawned pairs, hatched in the laboratory, were transplanted to replicate anemones at three replicate ALAN and 3 replicate control sites. Repeated observations of survivorship (presence/absence) and specific growth rate were made over the experimental period. This study is an important advancement in understanding the effects of ALAN in tropical marine environments, specifically on coral reef fishes, as it tests key demographic traits at replicate ALAN and control sites under natural conditions of predation and food supply. The experiment is expertly designed and the results appropriately analysed with linear mixed-effects models. Results are sensibly presented and interpreted. The text is very cleanly written and easy to follow. Overall, an excellent study that provides novel insight to the effects of ALAN on reef fishes in the wild.

I have just two minor comments regarding interpretation. First, I don't think any conclusions can be drawn about the possible effects of ALAN on fish recruitment based on just one new fish recruiting over the entire study population of anemones. Sure, it shows that ALAN does not totally preclude anemonefish settlement, but with only one fish recruiting it's impossible to say anything about relative effects. I think this paragraph (lines 313-323) should probably be deleted as it's entirely speculative.

In place of this paragraph it would be worth noting that anemonefish usually settle to anemones where other anemonefish are already present, and this can subsequently influence their growth rate through the social control of growth. Mortality rates of new recruits can also be modulated by the presence of larger individuals, which dominate prime space and may force juveniles to use more exposed locations on the anemone. In this study it has been necessary to simplify the social organization to a single juvenile in order to conduct such a robust experiment, but as a result the patterns of growth and survival may not be entirely representative of anemonefishes in natural social groups when exposed to ALAN. Some comment on this is warranted in the discussion.

Review form: Reviewer 2 (Natalie Roos)

Recommendation

Accept with minor revision (please list in comments)

It is a condition of publication that authors make their supporting data, code and materials available - either as supplementary material or hosted in an external repository. Please rate, if applicable, the supporting data on the following criteria.

Is it accessible?

No

Is it clear?

N/A

Is it adequate?

N/A

Do you have any concerns about statistical analyses in this paper? If so, please specify them explicitly in your report.

No

Comments to the Author

Major comment

This is an important contribution investigating the effects of long-term exposure of ALAN on the growth and survival of a reef fish. Data is well analyzed and the text is well written. The manuscript is concisely summarized with informative figures, and the modeling results add insights. I believe that the manuscript can be improved with some (not too much) restructuring and more clarification. My main concern is: other anthropogenic stressors that may affect anemonefish were not investigated or were poorly discussed. Nearshore environments have many other stressors rather than only ALAN (sound pollution was briefly approached, yet it may be a major stressor, see Leduc et al. 2021 Biological Conservation). For example, northern lagoons of Moorea have higher levels of nitrogen pollution nearshore, making some scleractinian corals more vulnerable to heat stress (please see Donovan et al. 2020 PNAS). Even though anemones may respond differently to nitrogen pollution, it is important to stress that, by the end of the experiment, 22.22% (n = 4) of the anemones were absent in the ALAN sites, against 12.5% (n = 3) in the controls. Authors should discuss possible stressors affecting the hosts.

Please see below specific comments.

Figure 1. It would be worth adding the map of Moorea showing the three ALAN sites and controls. It is not clear whether the control sites were further from the coast than the ALAN sites. This would help to understand the spatial setting of the experiment. Please mention the distance from the coast of each site.

Lines 138 – 142. Please better describe how the fieldwork was conducted.

Lines 147. What were the distances between anemones?

Line 183. LMERS. Please introduce the acronym before using it in the text.

Line 257. Please mention that *Drosophila melanogaster* is a fruit fly.

Decision letter (RSPB-2021-0454.R0)

12-Apr-2021

Dear Dr Mills:

Your manuscript has now been peer reviewed and the reviews have been assessed by an Associate Editor. The reviewers' comments (not including confidential comments to the Editor) and the comments from the Associate Editor are included at the end of this email for your reference. As you will see, the reviewers and the Editors have raised some concerns with your manuscript and we would like to invite you to revise your manuscript to address them.

Research ethics:

Use of animals and field studies:

It is a condition of publication that you make available the data and research materials supporting the results in the article. Please see our Data Sharing Policies (<https://royalsociety.org/journals/authors/author-guidelines/#data>). Datasets should be deposited in an appropriate publicly available repository and details of the associated accession number, link or DOI to the datasets must be included in the Data Accessibility section of the article (<https://royalsociety.org/journals/ethics-policies/data-sharing-mining/>). Reference(s) to datasets should also be included in the reference list of the article with DOIs (where available).

Please submit a copy of your revised paper within three weeks. If we do not hear from you within this time your manuscript will be rejected. If you are unable to meet this deadline please let us know as soon as possible, as we may be able to grant a short extension.

Best wishes,
Dr Daniel Costa
mailto:proceedingsb@royalsociety.org

Associate Editor
Comments to Author:

We have obtained two expert reviews of your manuscript and both were supportive of your study and I agree this paper could be publishable in Proceedings B. However, both referees provided suggestions for improvement including important factors that should be addressed: the potential for confounding factors that could have influenced your results including fish survival (and possibly affected the host anemones), such as nitrogen pollution, and sound pollution could be expanded in the manuscript as well. These possibilities need to be carefully considered and discussed, especially given the experiment took place in the wild - were these potential factors controlled for, or were they a concern? Were there potentially other hypotheses that could have explained or contributed to your results? Furthermore, a discussion about the biology of the fish would be beneficial, as social controls also influence growth in the species. My read of the reviews and the paper suggests that these factors need to be taken into account to convince readers that the most likely reason for the lower survival and growth rates of anemonefish in this study is, indeed, ALAN and nothing else.

I also would like to note that the link to the DRYAD repository appears to be broken.

Reviewer(s)' Comments to Author:
Referee: 1

Comments to the Author(s)

This novel study uses a field-based experiment to test the effects of artificial light at night (ALAN) on the growth and survival of juvenile anemonefishes over a ~20 month period. Juvenile

anemonefish from wild-spawned pairs, hatched in the laboratory, were transplanted to replicate anemones at three replicate ALAN and 3 replicate control sites. Repeated observations of survivorship (presence/absence) and specific growth rate were made over the experimental period. This study is an important advancement in understanding the effects of ALAN in tropical marine environments, specifically on coral reef fishes, as it tests key demographic traits at replicate ALAN and control sites under natural conditions of predation and food supply. The experiment is expertly designed and the results appropriately analysed with linear mixed-effects models. Results are sensibly presented and interpreted. The text is very cleanly written and easy to follow. Overall, an excellent study that provides novel insight to the effects of ALAN on reef fishes in the wild.

I have just two minor comments regarding interpretation. First, I don't think any conclusions can be drawn about the possible effects of ALAN on fish recruitment based on just one new fish recruiting over the entire study population of anemones. Sure, it shows that ALAN does not totally preclude anemonefish settlement, but with only one fish recruiting it's impossible to say anything about relative effects. I think this paragraph (lines 313-323) should probably be deleted as it's entirely speculative.

In place of this paragraph it would be worth noting that anemonefish usually settle to anemones where other anemonefish are already present, and this can subsequently influence their growth rate through the social control of growth. Mortality rates of new recruits can also be modulated by the presence of larger individuals, which dominate prime space and may force juveniles to use more exposed locations on the anemone. In this study it has been necessary to simplify the social organization to a single juvenile in order to conduct such a robust experiment, but as a result the patterns of growth and survival may not be entirely representative of anemonefishes in natural social groups when exposed to ALAN. Some comment on this is warranted in the discussion.

Referee: 2

Comments to the Author(s)

Major comment

This is an important contribution investigating the effects of long-term exposure of ALAN on the growth and survival of a reef fish. Data is well analyzed and the text is well written. The manuscript is concisely summarized with informative figures, and the modeling results add insights. I believe that the manuscript can be improved with some (not too much) restructuring and more clarification. My main concern is: other anthropogenic stressors that may affect anemonefish were not investigated or were poorly discussed. Nearshore environments have many other stressors rather than only ALAN (sound pollution was briefly approached, yet it may be a major stressor, see Leduc et al. 2021 Biological Conservation). For example, northern lagoons of Moorea have higher levels of nitrogen pollution nearshore, making some scleractinian corals more vulnerable to heat stress (please see Donovan et al. 2020 PNAS). Even though anemones may respond differently to nitrogen pollution, it is important to stress that, by the end of the experiment, 22.22% (n = 4) of the anemones were absent in the ALAN sites, against 12.5% (n = 3) in the controls. Authors should discuss possible stressors affecting the hosts. Please see below specific comments.

Figure 1. It would be worth adding the map of Moorea showing the three ALAN sites and controls. It is not clear whether the control sites were further from the coast than the ALAN sites. This would help to understand the spatial setting of the experiment. Please mention the distance from the coast of each site.

Lines 138 - 142. Please better describe how the fieldwork was conducted.

Lines 147. What were the distances between anemones?

Line 183. LMERS. Please introduce the acronym before using it in the text.

Line 257. Please mention that *Drosophila melanogaster* is a fruit fly.

Author's Response to Decision Letter for (RSPB-2021-0454.R0)

See Appendix A.

Decision letter (RSPB-2021-0454.R1)

25-Apr-2021

Dear Dr Mills

I am pleased to inform you that your Review manuscript RSPB-2021-0454.R1 entitled "Long-term exposure to artificial light at night (ALAN) in the wild decreases growth and survival of a coral reef fish" has been accepted for publication in Proceedings B.

The referee(s) do not recommend any further changes. Therefore, please proof-read your manuscript carefully and upload your final files for publication. Because the schedule for publication is very tight, it is a condition of publication that you submit the revised version of your manuscript within 7 days. If you do not think you will be able to meet this date please let me know immediately.

To upload your manuscript, log into <http://mc.manuscriptcentral.com/prsb> and enter your Author Centre, where you will find your manuscript title listed under "Manuscripts with Decisions." Under "Actions," click on "Create a Revision." Your manuscript number has been appended to denote a revision.

You will be unable to make your revisions on the originally submitted version of the manuscript. Instead, upload a new version through your Author Centre.

1) A text file of the manuscript (doc, txt, rtf or tex), including the references, tables (including captions) and figure captions. Please remove any tracked changes from the text before submission. PDF files are not an accepted format for the "Main Document".

2) A separate electronic file of each figure (tiff, EPS or print-quality PDF preferred). The format should be produced directly from original creation package, or original software format. Please note that PowerPoint files are not accepted.

3) Electronic supplementary material: this should be contained in a separate file from the main text and the file name should contain the author's name and journal name, e.g. `authorname_procb_ESM_figures.pdf`

All supplementary materials accompanying an accepted article will be treated as in their final form. They will be published alongside the paper on the journal website and posted on the online figshare repository. Files on figshare will be made available approximately one week before the accompanying article so that the supplementary material can be attributed a unique DOI. Please see: <https://royalsociety.org/journals/authors/author-guidelines/>

4) Data-Sharing and data citation

It is a condition of publication that data supporting your paper are made available. Data should be made available either in the electronic supplementary material or through an appropriate repository. Details of how to access data should be included in your paper. Please see <https://royalsociety.org/journals/ethics-policies/data-sharing-mining/> for more details.

If you wish to submit your data to Dryad (<http://datadryad.org/>) and have not already done so you can submit your data via this link <http://datadryad.org/submit?journalID=RSPB&manu=RSPB-2021-0454.R1> which will take you to your unique entry in the Dryad repository.

Once again, thank you for submitting your manuscript to Proceedings B and I look forward to receiving your final version. If you have any questions at all, please do not hesitate to get in touch.

Sincerely,
Dr Daniel Costa
Editor, Proceedings B
<mailto:proceedingsb@royalsociety.org>

Associate Editor Board Member

Comments to Author:

I am pleased to inform you that the revisions you have made as per the reviewers' comments have improved the manuscript and clarified concerns about confounding effects on the outcome of your study. However, in my reading there are a few more revisions that would improve the manuscript that need to be addressed expeditiously, mostly along the lines of further clarifications, and caution regarding interpretation of results, to ensure your study and its outcomes are fully contextualized. I have provided a few specific revisions I would like to see below.

Throughout: please consider revising to active voice.

Lines 89-90: Please make a note of how long the average lifespan of the anemone-fish is.

There is some confusion about how many species you are studying. Is it one species or six? In line 159 you mention "of the six species of anemonefish tested..." but prior to that, in Line 110 you mention "six anemonefish territories". I believe the mention of six species in Line 159 is in reference to another study, as it is referenced, but given the coincidence in the number six in both cases (number of territories in your study, and number of species in the other study), it would be worthwhile to clarify this with something in Line 159 like, "Previously [46] studied six anemonefish and found..."

Lines 231-233: OK, but the sites are barely different, so please be cautious with interpreting. Further, the p-value of your ALAN treatment was 0.055; not significant.

Lines 297-299: I disagree that ALAN had an "impact" on growth - height was the only thing that was statistically lower than the control group. Weight and length were not different, and further your sample sizes were very small. It seems to me that a solid reason for lower survival and not much affect on growth (on the animals that did survive) is that the extra light allowed predators to eat them, which I think gets not as much attention in your discussion section as it should. I am less convinced about the conversation around physiology and hormone responses and I am not sure that level of detail is warranted given the small sample size combined with the fact there was not a substantial impact of ALAN on any growth metric other than height. But I do think increased predation is a likely explanation and would like to see that as a primary (or more prominent) discussion point.

Line 334: Please define species name for crucian carp, and ensure all species names are defined in the first instance throughout.

Author's Response to Decision Letter for (RSPB-2021-0454.R1)

See Appendix B.

Decision letter (RSPB-2021-0454.R2)

17-May-2021

Dear Dr Mills

I am pleased to inform you that your manuscript entitled "Long-term exposure to artificial light at night (ALAN) in the wild decreases survival and growth of a coral reef fish" has been accepted for publication in Proceedings B.

Data Accessibility section

Open Access

Paper charges

Sincerely,

Dr Daniel Costa

Appendix A

MS Reference Number: RSPB-2021-0454

MS Title: **Long-term exposure to artificial light at night (ALAN) in the wild decreases growth and survival of a coral reef fish**

MS Authors: Schligler, Jules; Cortese, Daphne; Beldade, Ricardo; Swearer, Stephen E.; Mills, Suzanne

Response to reviewers

We are grateful to the reviewers for their positive and constructive comments. In addressing those (responses provided in blue beneath each comment with line numbers referring to the main resubmitted revised document), we believe we have strengthened the paper still further. At the end of the replies to reviewers we have included the revised manuscript with tracked changes highlighted in light blue, unfortunately without line numbers.

Associate Editor

Comments to Author:

We have obtained two expert reviews of your manuscript and both were supportive of your study and I agree this paper could be publishable in Proceedings B. However, both referees provided suggestions for improvement including important factors that should be addressed: the potential for confounding factors that could have influenced your results including fish survival (and possibly affected the host anemones), such as nitrogen pollution, and sound pollution could be expanded in the manuscript as well. These possibilities need to be carefully considered and discussed, especially given the experiment took place in the wild - were these potential factors controlled for, or were they a concern? Were there potentially other hypotheses that could have explained or contributed to your results? Furthermore, a discussion about the biology of the fish would be beneficial, as social controls also influence growth in the species. My read of the reviews and the paper suggests that these factors need to be taken into account to convince readers that the most likely reason for the lower survival and growth rates of anemonefish in this study is, indeed, ALAN and nothing else.

We have added a paragraph that includes the possible confounding factors suggested, but also by adding a map of our sample sites and distances from land, and discussing the information known about the different stressors around Moorea, we suggest that it is unlikely that the two treatments were exposed differently to anthropogenic stressors other than ALAN. We have also added a discussion about the biology of the fish, including the presence of social control on growth rates, but we have also included a discussion of recruitment in *Amphiprion* in general and how the study species used in this work differs from most other anemonefish species as in Moorea it lives in smaller social groups and larger territories (i.e. multiple anemones) and so may be less impacted by social control than other anemonefish species.

I also would like to note that the link to the DRYAD repository appears to be broken.

We thank you for pointing this out – we have deposited the data again in DRYAD.

Referee: 1

Comments to the Author(s)

This novel study uses a field-based experiment to test the effects of artificial light at night (ALAN) on the growth and survival of juvenile anemonefishes over a ~20 month period. Juvenile anemonefish from wild-spawned pairs, hatched in the laboratory, were transplanted to replicate anemones at three replicate ALAN and 3 replicate control sites. Repeated observations of survivorship (presence/absence) and specific growth rate were made over the experimental period. This study is an important advancement in understanding the effects of ALAN in tropical marine environments, specifically on coral reef fishes, as it tests key demographic traits at replicate ALAN and control sites under natural conditions of predation and food supply. The experiment is expertly designed and the results appropriately analysed with linear mixed-effects models. Results are sensibly presented and interpreted. The text is very cleanly written and easy to follow. Overall, an excellent study that provides novel insight to the effects of ALAN on reef fishes in the wild.

I have just two minor comments regarding interpretation. First, I don't think any conclusions can be drawn about the possible effects of ALAN on fish recruitment based on just one new fish recruiting over the entire study population of anemones. Sure, it shows that ALAN does not totally preclude anemonefish settlement, but with only one fish recruiting it's impossible to say anything about relative effects. I think this paragraph (lines 313-323) should probably be deleted as it's entirely speculative.

We agree that this is too speculative and have deleted this paragraph entirely.

In place of this paragraph it would be worth noting that anemonefish usually settle to anemones where other anemonefish are already present, and this can subsequently influence their growth rate through the social control of growth. Mortality rates of new recruits can also be modulated by the presence of larger individuals, which dominate prime space and may force juveniles to use more exposed locations on the anemone. In this study it has been necessary to simplify the social organization to a single juvenile in order to conduct such a robust experiment, but as a result the patterns of growth and survival may not be entirely representative of anemonefishes in natural social groups when exposed to ALAN. Some comment on this is warranted in the discussion.

We agree that the literature shows that conspecific presence impacts growth and mortality of anemonefish settlers. We have added a paragraph in the discussion introducing these points, but also a short description of our study species and how this may impact the results of our study on lines 340-352.

Referee: 2

Comments to the Author(s)

Major comment

This is an important contribution investigating the effects of long-term exposure of ALAN on the growth and survival of a reef fish. Data is well analyzed and the text is well written. The manuscript is concisely summarized with informative figures, and the modeling results add insights.

We thank the reviewer for these comments.

I believe that the manuscript can be improved with some (not too much) restructuring and more clarification. My main concern is: other anthropogenic stressors that may affect anemonefish were not investigated or were poorly discussed. Nearshore environments have many other stressors rather than only ALAN (sound pollution was briefly approached, yet it may be a major stressor, see Leduc et al. 2021 Biological Conservation). For example, northern lagoons of Moorea have higher levels of nitrogen pollution nearshore, making some scleractinian corals more vulnerable to heat stress (please see Donovan et al. 2020 PNAS). Even though anemones may respond differently to nitrogen pollution, it is important to stress that, by the end of the experiment, 22.22% (n = 4) of the anemones were absent in the ALAN sites, against 12.5% (n = 3) in the controls. Authors should discuss possible stressors affecting the hosts.

We thank the reviewer for these points which were interesting, that we hadn't previously considered and which we have now added to the Discussion. However, we also realise that we hadn't emphasised sufficiently how close our control and ALAN sites were to each other and to land – we have now rectified this in the text and in Figure S1. We chose our design specifically to incorporate six replicates of each treatment at three different Sites around the island in order to remove any other confounding effects of Site, such as other anthropogenic stressors. Of the anthropogenic stressors that the reviewer mentioned (nitrogen pollution and land-based sound pollution) we have also added marine-based sound pollution as well as human disturbance and whilst indeed these four stressors may have impacted the ALAN and Control territories differently at each site (see sections below), we show/predict that they would never impact the treatments in the same way across all three sites. Based on this, we believe our experimental design took into account differences in these potential other anthropogenic stressors and yet we still found a significant effect of ALAN, hence we have added and discuss these points briefly in the manuscript, but not at length, as we also lack any quantitative measures to test different the hypotheses accurately. We explain our reasoning below.

Nitrogen pollution

Between treatments: Leduc et al (2021) measured nitrogen pollution in Moorea lagoon (Figure 2 from Leduc et al 2021 below). We have added white boxes (on right) to represent the locations of our three Sites (Figure S1). At Sites 2 and 3, nitrogen levels do not differ within the white box (i.e. within Site) and hence will not differ between treatments within each site regardless of their distance from each other. At Site 1 there is a small gradient in nitrogen levels with proximity to shore, but it actually increases, not decreases, further from shore, but Site 1 was the one site where our treatments were evenly distributed in terms of distance from land and from each other (Figure.

S1). Therefore, we do not believe that there are any differences in nitrogen pollution between treatments, and hence nitrogen pollution would not impact our main results.

Between Sites: Based on Ludec et al (2021) we predict that nitrogen pollution would differ between sites increasing in the following order: Site 2 < Site 3 < Site 1. In terms of results, anemonefish survival and growth was highest at Site 2 compared to Site 3 and Site 1. The trends are similar and would be interesting to follow up with quantitative data in the future, but not sufficient to warrant a lengthy discussion in the manuscript.

Sound pollution

Between treatments: Land-based sound pollution would decrease with distance from the coast, therefore the second Control territories at Site 2 and the Control territories at Site 3 would indeed be less impacted than ALAN territories at the same Site. On the other hand, ALAN and Control territories at Site 1 and the first Control site at Site 2 are a similar distance from land. However, marine-based sound pollution due to motorised vessel noise would impact territories in the opposite way – increasing with distance from land and proximity to boat channels. Therefore, the Control territories further from land at Site 2 and Site 3 are closer to boat channels and as such would be more impacted by marine-based rather than land-based sound pollution. Therefore, when considered as a whole, it is unlikely that differences in sound pollution between treatments, land- and marine-based combined, would impact our results.

Between Sites: Based on knowledge of boat channel use around Moorea (pers comm) we predict that marine-based sound pollution would differ between sites increasing in the following order: Site 3 < Site 1 < Site 2. Based on size of hotel resort which is given in Figure S1 (Site 1 = 94 guest rooms; Site 2 = 245 guest rooms; Site 3 = 113 guest rooms), we predict that land-based sound pollution would differ between sites increasing in the following order: Site 1 < Site 3 < Site 2. In terms of results, anemonefish survival and growth were highest at Site 2, which we have predicted to the highest land- and marine-based sound pollution, so we do not believe our results are confounded by sound pollution.

Human disturbance

Between treatments: Human disturbance at our territories would be related to distance from land, decreasing with distance from land, therefore the second Control territories at Site 2 would indeed be less impacted by human disturbance than ALAN territories, but this is the only Site. Control territories at Site 3 are further from land (Figure S1), but they are closer to the hotel resort bungalows (which each have a ladder into the lagoon) and human disturbance. Therefore, we do not believe that one Control site with lower human disturbance would be driving our results between treatments.

Between Sites: We predict that land-based tourism-related human disturbance would correlate with size of hotel resort which is given in Figure S1 (Site 1 = 94 guest rooms; Site 2 = 245 guest rooms; Site 3 = 113 guest rooms), and would differ between sites increasing in the following order: Site 1 < Site 3 < Site 2. In terms of results, anemonefish survival and growth was highest at Site 2 compared to Site 3 and Site 1, Site 2 is likely the most impacted by human disturbance, therefore we do not believe our results are confounded by human disturbance.

On a separate note, we also believe we misled the reviewer into thinking that $n = 4$ and $n = 3$ anemones were absent at the end of the experiment – the real figures are 7 and 7, or 29.1% and 38.8% mortality for Controls and ALAN respectively. We have added this to the manuscript (lines 171-175).

Please see below specific comments.

Figure 1. It would be worth adding the map of Moorea showing the three ALAN sites and controls. It is not clear whether the control sites were further from the coast than the ALAN sites. This would help to understand the spatial setting of the experiment. Please mention the distance from the coast of each site.

We have added a map in Suppl material as Figure S1 and we have added distance from coast at each site in the main text.

Lines 138 – 142. Please better describe how the fieldwork was conducted.

We have now added more detail concerning how the monitoring was carried out (lines 147-150).

Lines 147. What were the distances between anemones?

We have now added distances between anemones within and between treatments (lines 155-157).

Line 183. LMERS. Please introduce the acronym before using it in the text.

We have now introduced the acronym on line 189.

Line 257. Please mention that *Drosophila melanogaster* is a fruit fly.

We have now added fruit fly as requested on line 272.

.....

Appendix B

MS Reference Number: RSPB-2021-0454

MS Title: **Long-term exposure to artificial light at night (ALAN) in the wild decreases growth and survival of a coral reef fish**

MS Authors: Schligler, Jules; Cortese, Daphne; Beldade, Ricardo; Swearer, Stephen E.; Mills, Suzanne

Replies to Associate Editor

Associate Editor Board Member

Comments to Author:

I am pleased to inform you that the revisions you have made as per the reviewers' comments have improved the manuscript and clarified concerns about confounding effects on the outcome of your study. However, in my reading there are a few more revisions that would improve the manuscript that need to be addressed expeditiously, mostly along the lines of further clarifications, and caution regarding interpretation of results, to ensure your study and its outcomes are fully contextualized. I have provided a few specific revisions I would like to see below.

We thank the Editor for the additional comments and we have made changes to the text which are highlighted in yellow.

Throughout: please consider revising to active voice.

The method section is now completely in the active voice.

Lines 89-90: Please make a note of how long the average lifespan of the anemone-fish is.

We have added this information on lines 90-91.

There is some confusion about how many species you are studying. Is it one species or six? In line 159 you mention "of the six species of anemonefish tested..." but prior to that, in Line 110 you mention "six anemonefish territories". I believe the mention of six species in Line 159 is in reference to another study, as it is referenced, but given the coincidence in the number six in both cases (number of territories in your study, and number of species in the other study), it would be worthwhile to clarify this with something in Line 159 like, "Previously [46] studied six anemonefish and found..."

Indeed this could be confusing. This has been changed to "Nevertheless, a previous study testing movement behaviour in six anemonefish species found that adult and juvenile A. chrysopterus were the most likely to move (13% changed host anemone within 4 months compared to 6% in A. clarkii and 0% in four other Amphiprion species)[46].", on lines 160-163.

Lines 231-233: OK, but the sites are barely different, so please be cautious with interpreting. Further, the p-value of your ALAN treatment was 0.055; not significant.

We agree, we make sure not to say that the difference is significant and now discuss the presence of a tendency for growth to be lower on lines 226-229.

Lines 297-299: I disagree that ALAN had an "impact" on growth - height was the only thing that was statistically lower than the control group. Weight and length were not different, and further your sample sizes were very small. It seems to me that a solid reason for lower survival and not much affect on growth (on the animals that did survive) is that the extra light allowed predators to eat them, which I think gets not as much attention in your discussion section as it should. I am less convinced about the conversation around physiology and

hormone responses and I am not sure that level of detail is warranted given the small sample size combined with the fact there was not a substantial impact of ALAN on any growth metric other than height. But I do think increased predation is a likely explanation and would like to see that as a primary (or more prominent) discussion point.

We have now switched the order in which we discuss the potential causes of lower survival and discuss predation first, in order to give it more weight. However as lower survival has been found in laboratory studies in the absence of predators (references in the manuscript), then we prefer to maintain our short discussion on the physiological impacts of ALAN in addition to predation, but we have slightly reduced the discussion of factors other than predation.

However, whilst height was the only growth measure statistically lower (15%) under ALAN, growth in terms of weight showed a NS trend ($p = 0.07$; 51% lower; Fig. 3) and length was 21% lower under ALAN (although not statistically significant likely due to the low sample size of surviving individuals). Therefore, we are convinced that our results are relevant and would have been more significant had more individuals survived.

We have also switched the order to survival and growth in the title and in the abstract to emphasise the survival result over that of growth.

Line 334: Please define species name for crucian carp, and ensure all species names are defined in the first instance throughout.

Checked and added as requested.